# Endothelial Dysfunction Accelerates Impairment of Mitochondrial Function in Ageing Kidneys via Inflammasome Activation

**DOI:** 10.3390/ijms22179269

**Published:** 2021-08-27

**Authors:** Yoshihisa Wada, Reina Umeno, Hajime Nagasu, Megumi Kondo, Atsuyuki Tokuyama, Hiroyuki Kadoya, Kengo Kidokoro, Shun’ichiro Taniguchi, Masafumi Takahashi, Tamaki Sasaki, Naoki Kashihara

**Affiliations:** 1Department of Nephrology and Hypertension, Kawasaki Medical School, 577 Matsushima, Kurashiki 701-0192, Okayama, Japan; y.wada.1017@med.kawasaki-m.ac.jp (Y.W.); m0110012g@gmail.com (R.U.); mgg530@gmail.com (M.K.); t.yama730@gmail.com (A.T.); kadoya-hiro@med.kawasaki-m.ac.jp (H.K.); k.kid@med.kawasaki-m.ac.jp (K.K.); tsasaki@med.kawasaki-m.ac.jp (T.S.); kashinao@med.kawasaki-m.ac.jp (N.K.); 2Department of Hematology and Medical Oncology, Shinshu University Graduate School of Medicine, Matsumoto 390-8621, Nagano, Japan; stangch@shinshu-u.ac.jp; 3Division of Inflammation Research, Center for Molecular Medicine, Jichi Medical University, Shimotsuke 329-0498, Tochigi, Japan; masafumi2@jichi.ac.jp

**Keywords:** inflammasome, senescence-associated secretory phenotype, macrophages, mitochondria dysfunction, ASC-deficient mice

## Abstract

Chronic kidney disease is a common problem in the elderly and is associated with increased mortality. We have reported on the role of nitric oxide, which is generated from endothelial nitric oxide synthase (eNOS), in the progression of aged kidneys. To elucidate the role of endothelial dysfunction and the lack of an eNOS-NO pathway in ageing kidneys, we conducted experiments using eNOS and ASC-deficient mice. C57B/6 J mice (wild type (WT)), *eNOS* knockout (eNOS KO), and *ASC* knockout (ASC KO) mice were used in the present study. Then, eNOS/ASC double-knockout (eNOS/ASC DKO) mice were generated by crossing eNOS KO and ASC KO mice. These mice were sacrificed at 17−19 months old. The Masson positive area and the KIM-1 positive area tended to increase in eNOS KO mice, compared with WT mice, but not eNOS/ASC DKO mice. The COX-positive area was significantly reduced in eNOS KO mice, compared with WT and eNOS/ASC DKO mice. To determine whether inflammasomes were activated in infiltrating macrophages, the double staining of IL-18 and F4/80 was performed. IL-18 and F4/80 were found to be co-localised in the tubulointerstitial areas. Inflammasomes play a pivotal role in inflammaging in ageing kidneys. Furthermore, inflammasome activation may accelerate cellular senescence via mitochondrial dysfunction. The importance of endothelial function as a regulatory mechanism suggests that protection of endothelial function may be a potential therapeutic target.

## 1. Introduction

Chronic kidney disease (CKD) is a common problem in the elderly and is associated with increased mortality. Age-related vascular changes are observed in ageing-related nephrosclerosis. Arteriosclerosis-related renal morphological changes, such as an increased number of hyalinised glomeruli, a decreased number of total glomeruli, tubulointerstitial fibrosis, and fibrous thickening of the vascular endothelium, are increased with ageing. These morphological changes are associated with endothelial dysfunction. A decrease of the nitric oxide (NO), which is generated from endothelial nitric oxide synthase (eNOS), has been observed in the progression of aged kidneys. L-NG-nitroarginine methyl ester (L-NAME) treatment increases angiostatin via the inhibition of Cathepsin D, resulting in interstitial fibrosis [1]. In fact, the vascular dysfunction caused by endothelial dysfunction is one of the main vasculature alterations during cardiovascular ageing [2,3]. In addition, endothelial dysfunction, which is assessed by flow-medicated vasodilatation, is observed in CKD patients [4]. The reduced abundance or activity of eNOS are also caused by the increased levels of endogenous NOS inhibitors. Arginase was in contact with NOS, contributing to impaired relaxation of the endothelium. In brief, arginase expression and activity increase in the vasculature with advancing age and leads to endothelial dysfunction [5]. One of the mechanisms of impairment of the eNOS-NO pathway is the production of reactive oxidative species (ROS). ROS are critical determinants of the bioavailability of NO. Moreover, ROS scavenge NO, resulting in the generation of peroxynitrite, which further decreases NO bioavailability [6]. NO production is impaired in both CKD [7] and the ageing kidney [8], coinciding with the progression of renal injury and decreased renal plasma flow. These facts suggested that endothelial dysfunction, that is, the lack of an eNOS-NO pathway, accelerates ageing-related kidney dysfunction.

Chronic inflammation is one of the common pathways in the progression of kidney diseases. Especially in the nucleotide-binding domain, the leucine-rich repeat-containing protein (NLRP3) inflammasome plays a pivotal role in chronic inflammation. Inflammasomes are multimeric protein complexes that form in the cytosol in response to either exogenous pathogens or endogenous danger signals and induce proinflammatory effects. The importance of inflammasome activation in the progression of kidney disease has been well known [9,10]. The inhibition of caspase 1 activation could reduce the infiltration of pro-inflammatory macrophages, called M1 macrophages, in unilateral ureter obstruction [11]. Inflammasome activation is important for prolonging inflammation in kidney diseases, resulting in fibrotic change.

Recently, chronic inflammation with ageing has been called ‘inflammaging’. Inflammaging is chronic inflammation associated with the development of ageing-related organ damage. It is well understood that inflammation occurs due to cellular senescence. Senescent cells secrete a pro-inflammatory factor called the ‘senescence-associated secretory phenotype’ (SASP). The expression of the SASP is regulated by inflammasome activation [12]. Caspase 1 activation leads to pyroptosis, which is caspase 1-dependent programmed cell death, resulting in SASP releasing [13]. Pyroptosis is closely associated with the development of atherosclerosis, referring to pyroptosis in endothelial cells and macrophages [14]. The NLRP3 inflammasome is one of the major sources of inflammation in age-related diseases [15]. Several types of inflammasome were notably activated in the ageing kidneys of rats [16]. However, the significance of the inflammasome in age-related renal injury is not fully understood. Especially the relationship between inflammasome-related inflammation and the eNOS-NO pathway has not been elucidated.

## 2. Results

### 2.1. Inflammatory Response Worse in Aged eNOS KO Mice, Causing Glomerular Damage

The systolic blood pressure in *eNOS* knockout mice (eNOS KO) and *eNOS/ASC* double-knockout (eNOS/ASC DKO) mice was significantly higher than that in C57B/6 J (wild type (WT)) mice (Figure 1a). Furthermore, there were significant differences in SBP between eNOS/ASC DKO mice and eNOS KO mice (Figure 1a). Serum creatinine was significantly elevated in eNOS KO mice, compared with WT mice (Figure 1b). Body weight was significantly lower in eNOS KO and eNOS/ASC DKO mice than WT mice. There was no difference in body weight between eNOS KO and eNOS/ASC DKO mice (Figure 1c). The percentage of glomerular sclerosis in eNOS KO mice was increased, compared to WT mice (Figure 2a,b). Urinary albumin excretion was significantly increased in eNOS KO mice, compared with WT mice. These glomerular injuries improved in eNOS/ASC DKO mice. The next inflammatory response in glomeruli was assessed by the mRNA levels of the inflammasome and fibrosis-related gene (Figure 2d–h). ASC, IL18, and IL6, which are inflammasome components and related cytokines, as well as mRNA expression in glomeruli of eNOS KO mice, increased compared with WT but not eNOS/ASC DKO mice. These results suggest that eNOS-NO deficiency may accelerate age-related glomerular damage and exacerbate inflammation, which is related to the inflammasome in glomeruli.

### 2.2. Lacking eNOS-NO Pathway Hastens Mitochondrial Damage and Cellular Senescence

Tubular interstitial fibrosis was evaluated by Masson trichrome staining, and tubular cell damage was assessed by kidney injury molecule 1 (KIM-1) staining. The Masson positive area and the KIM-1 positive area tended to increase in eNOS KO mice but not eNOS/ASC DKO mice (Figure 3a–c). Changes in the respiratory chain function, an indicator of mitochondrial damage, were assessed by the histochemical measurement of cytochrome c oxidase (COX), which is complex IV, and succinate dehydrogenase (SDH), which is complex II, activities. COX is a large transmembrane protein that functions as a terminal acceptor of the electron transport chain. It is one of the most important enzymes regulating mitochondrial function. Conversely, SDH is encoded by nuclear DNA, and its activity is usually not affected by mitochondrial DNA damage. The COX-positive area was significantly reduced in eNOS KO mice, compared with WT and eNOS/ASC DKO mice (Figure 3d,e). SDH staining did not significantly change in each group. It is well known that senescent cells show the accumulation of β-galactosidase; thus, senescence-associated β-galactosidase (SA β-gal) is useful for determining cell senescence. SA The β-gal positive area was significantly elevated in eNOS KO mice, compared with WT and eNOS/ASC DKO mice (Figure 3f,g).

### 2.3. Aged-eNOS KO Mice Exhibit Impaired Fatty Acid Metabolism Associated with Mitochondrial Dysfunction in the Proximal Tubule

A whirl-like structure could be observed in tubules with mitochondrial dysfunction [17]. The observation of proximal tubules by transmission electron microscopy showed multiple whirl-like structures in aged-eNOS KO mice (Figure 4a,b). Aged-eNOS KO mice were thought to have an abnormal fatty acid metabolism associated with mitochondrial dysfunction in the proximal tubule. To determine whether inflammasomes were activated in infiltrating macrophages, the double staining of IL-18 and F4/80 was performed. In each group, macrophages were found in the tubulointerstitial areas, while eNOS KO mice showed a more extensive distribution (Figure 5a). IL-18 and F4/80 were found to be co-localised in the tubulointerstitial areas of eNOS KO mice (Figure 5b).

### 2.4. NO Suppresses the Activation of Inflammasomes in Macrophages

Using in vitro primary cultures of bone-marrow-derived macrophages (BMDMs), we examined the effect of the absence of eNOS on macrophage inflammasome activation. After incubation with lipopolysaccharide (LPS), the BMDMs were treated with ATP to activate the NLRP3 inflammasome. S-nitrosoglutathione (GSNO) was used as the NO donor. LPS + ATP stimulation significantly increased inflammasome-related cytokines (IL-6 and IL-1β) in the supernatant (Figure 6a,b), but GSNO treatment suppressed LPS + ATP stimulation of the IL-1β secretion into the supernatant (Figure 6b).

## 3. Discussion

The eNOS KO mice showed exacerbated glomerular injury with significantly increased serum creatinine, a percentage of glomerular sclerosis, urinary albumin excretion, and inflammasome-associated cytokine gene expression. In addition, eNOS KO mice showed a significant increase in COX and SA β-gal positive areas in tubular cells, and transmission electron microscopy revealed mitochondria morphological changes. These results suggest that aged-eNOS KO mice exacerbate the inflammatory response and accelerate mitochondrial dysfunction and cellular senescence, resulting in glomerular sclerosis and tubular cell injury. Furthermore, IL-18 was expressed in macrophages, which are F4/80 positive cells, in the tubular interstitium of aged-eNOS KO mice. NO donors suppressed the secretion of inflammasome-related cytokines in macrophages. These results suggest that endothelial dysfunction exacerbates kidney injury via the activation of inflammasomes on macrophages in ageing kidneys.

Ageing kidneys are known to cause renal tubular damage due to ischemia. In particular, fibrosis occurs mainly at the corticomedullary border, resulting in decreased renal dysfunction. In this study, we found that eNOS KO mice exacerbated interstitial fibrosis and tubular injury, as indicated by KIM1-positive cells. A possible mechanism for this is the exacerbation of ischemia due to endothelial dysfunction.

Previous reports revealed that mitochondrial dysfunction was exacerbated in eNOS KO mice. In eNOS KO mice, mitochondrial morphological changes in podocytes were observed by electron microscopy [18]. Decreased pyruvate kinase M2 (PKM2) activity is one mechanism that may explain the impairment of eNOS function and mitochondrial damage. In diabetic nephropathy, mitochondrial dysfunction is exacerbated by decreased PKM2 activity in podocytes [19]. It is also possible that these mitochondrial disorders exacerbated age-related organ damage. Mitochondrial dysfunction is considered a common pathway for the progression of renal disease. In substance, mitochondrial dysfunction has been reported to be important in many renal disease models [20,21]. We have reported that the mitochondrial dysfunction-induced impairment of angiogenesis in ageing kidneys results in hypoxia [22]. In particular, mitochondrial dysfunction activates the inflammasome via mitochondrial ROS. The NRLP3 inflammasome has been shown to cluster components near mitochondria upon activation [23]. It has been shown that mitochondrial ROS is essential for inflammasome activation, and when mitochondrial ROS is scavenged by mitQ, inflammasome activation does not occur. These facts suggest that endothelial dysfunction may have exacerbated mitochondrial dysfunction and led to inflammasome activation.

However, since this study used general ASC KO mice, we could not examine which cells are important for inflammasome activation in the ageing kidney. Inflammatory cells, mainly macrophages, are important cells that cause inflammasome activation. In a model of tubulointerstitial injury in which aldosterone activates inflammasomes, we found co-localisation of F4/80 and IL-18. Infiltrating macrophages have been shown to be important in the development of chronic tubulointerstitial inflammation and fibrosis [10]. The NLRP3 inflammasome is also involved in the formation of atherosclerosis [24].

Meanwhile, the importance of inflammasome activation in podocytes in glomerular lesions has also been raised. Thioredoxin interacting protein plays a pivotal role in inflammasome activation in podocytes [25]. The activation of inflammasomes in podocytes is also important in glomerular lesions of diabetic kidney disease [26,27]. In addition, there are some reports of inflammasome activation in tubular cells [28,29]. In the future, it will be necessary to use cell-specific genetically modified animals to clarify this issue, or bone marrow transplantation experiments could help to dissolve this issue. Recently, the significance of another type of inflammasome, AIM2 inflammasome activation, has also been reported. It has been shown that AIM2 activation in macrophages is important for CKD lesion development [30]. The ASC KO mice used in this study do not cause AIM2 inflammasome activation either; therefore, it is possible that AIM2 inflammasome activation is involved.

As mentioned above, there is a close relationship between inflammasome activation and mitochondrial dysfunction. While mitochondrial ROS is important for inflammasome activation, it has also been shown that inflammasome activation leads to mitochondrial dysfunction: Caspase 1 activation causes a decrease in mitochondrial membrane potential and an increase in the production of mitochondrial ROS [31,32]. Furthermore, it has been reported that inflammasome activation impairs autophagy. In particular, we have found the caspase 1 activation-dependent impairment of mitophagy is mediated by the cleavage of parkin [31]. The present study suggests that mitochondrial damage in the tubules is reduced in eNOS/ASC DKO mice, which may be due to the regulation of inflammasome activation.

Several papers have reported on the modulation of inflammasomes by NO. Endogenous NO derived from iNOS (an inducible form of NO synthase) also negatively regulates NLRP3 inflammasome activation. The depletion of iNOS resulted in the increased accumulation of dysfunctional mitochondria in response to LPS and ATP, which was responsible for the increased IL-1β production and caspase 1 activation [33]. NO generated from iNOS inhibits the processing of IL-1β, but the mechanism by which these molecules suppress inflammation may extend beyond simply inhibiting the activity of inflammasomes [34]. These papers focus on the effects of iNOS, but few studies have focused on the eNOS-NO pathway. In a previous report, NO can suppress Toll-like receptor 4 (TLR4) signalling in macrophages [11]. TLR4 signalling is important in tubulointerstitial inflammation and glomerular damage of diabetic kidney disease [35]. We found increased macrophage infiltration and co-localisation with IL-18 in tubulointerstitial tissues of eNOS KO mice by fluorescent antibody assay. LPS + ATP stimulation of BMDMs increased IL-6 and IL-1β in the supernatant, which were suppressed by NO donor. These results suggest that endothelium-derived NO may regulate the NLRP3 inflammasome in macrophages directly.

In this study, we also examined the survival rate and found that eNOS/ASC DKO mice were significantly worse than eNOS KO mice, perhaps due to the higher blood pressure in eNOS/ASC DKO mice (data not shown). These results indicate that endothelial dysfunction leads to the progression of age-related organ damage through the disruption of inflammasome activity regulation. Inflammasome plays a pivotal role in the inflammaging in ageing kidneys. Furthermore, inflammasome activation may accelerate cellular senescence via mitochondrial dysfunction and DAMPs (Figure 7). The molecular mechanism remains unclear, and further studies are needed.

## 4. Materials and Methods

### 4.1. Animal

The experimental protocols followed in the current study were approved by the Ethics Review Committee for Animal Experimentation at Kawasaki Medical School, Kurashiki, Japan (19-091; 15 January 2020). WT and eNOS KO mice were purchased from the Jackson Laboratory (Bar Harbor, ME) [36]. ASC KO mice were kindly provided by Takahashi M (Jichi Medical University, Shimotsuke, Japan) [37]. Subsequently, eNOS/ASC DKO mice were generated by intercrossing eNOS KO and ASC KO mice. The mice were housed in a temperature- and humidity-controlled room with a 14:10 h light–dark cycle and were fed standard laboratory animal chow with free access to tap water. All mice in each group (*n* = 3–6, male) were euthanised at 17–19 months of age, and blood and kidney tissue were collected. At the beginning of the experiment (8 weeks old), the mice weighed 22–28 g. Blood samples were collected by inserting a 21-gauge needle into the right atrium of the mouse. Kidneys were harvested from the body after perfusion with saline (15 mL) from the left atrium.

### 4.2. Physiological and Biochemical Measurements

Body weights were recorded, and blood pressures and pulse rates were measured using the tail-cuff method with an automatic sphygmomanometer (BP98A; Softron Co., Ltd., Tokyo, Japan). The 24 h urine samples were collected on the day before euthanasia. Urinary albumin levels were determined by ELISA using a murine microalbuminuria ELISA kit (Albuwell M; Exocell, Inc., Philadelphia, PA, USA). Urinary creatinine was measured using commercially available LabAssay Creatinine (Wako Pure Chemical Industries, Osaka, Japan). Serum creatinine levels were measured at a central clinical laboratory (SRL, Inc., Tokyo, Japan).

### 4.3. Histologic Analysis and Immunohistochemistry

Right kidney tissue was fixed in 4% paraformaldehyde and embedded in paraffin for histologic analysis. Tissue sections (2 μm thick) were deparaffinised and stained with periodic acid–Schiff (PAS) and Masson trichrome staining. Tissues were also processed for electron microscopy (Jem-1400; Jeol, Tokyo, Japan) to assess ultrastructural alterations in the tubule. Immunochemical staining was conducted on the deparaffinised kidney sections (4 μm thick) by heating them in a microwave at 500 W for 15 min for antigen retrieval, followed by incubating overnight with an antibody against KIM-1 (KCA031610A, R&D Systems, Minneapolis, MN, USA). The primary antibody was detected using the Histofine Simple Stain MAX-PO kit (Nichirei Corporation, Tokyo, Japan) and 3,3′-diaminobenzidine (Sigma-Aldrich; Merck KGaA, Darmstadt, Germany). Double immunofluorescence staining was performed on frozen cryostat sections (5 μm thick). Antibodies against F4/80 (MCA497GA; Bio-Rad Laboratories, Inc., Hercules, CA, USA) and IL-18 (PAB16177; Abnova, Taipei, Taiwan) were used. FITC-labelled anti-rabbit IgG (Dako, Agilent Technologies, Inc., Santa Clara, CA, USA) and Alexa Fluor 555-labelled antirat IgG (Cell Signaling Technology, Inc., Danvers, MA, USA) were used as secondary antibodies. The percentages of Masson blue–positive area and positive staining for KIM-1 were quantified using a colour image analyser (Keyence, Osaka, Japan). All global glomerular sclerosis were counted in each kidney section. The percentage of glomerular sclerosis was calculated as an assessment of glomerular damage.

### 4.4. Isolation of Glomeruli

The glomeruli were isolated by taking advantage of their preferential uptake of Dynabeads (M-450 Tosylactivated; Life Technologies Japan, Tokyo, Japan), as described previously [38], and processed for RNA extraction.

### 4.5. Real-Time Reverse Transcription PCR

Total RNA was isolated from isolated glomeruli with TRIzol (Thermo Fisher Scientific, Inc., Waltham, MA, USA) and digested using DNase (MilliporeSigma, Burlington, MA, USA). Total RNA was extracted using Phase Lock Gel Heavy 2 mL tubes (Quantabio, Beverly, MA, USA). Moloney murine leukaemia virus reverse transcriptase (Thermo Fisher Scientific, Inc.) was used to synthesise cDNA from total RNA (1 mg) using oligo(dT)_12–18_ as primers (Thermo Fisher Scientific, Inc.). Reverse transcription was performed for 50 min at 37 °C. The primers and probes for the TaqMan analysis were designed using sequence information from GenBank (National Institutes of Health, Bethesda, MD, USA) and Primer-BLAST online software (https://www.ncbi.nlm.nih.gov/tools/primer-blast/index.cgi, accessed on 25 August 2021).

The primer and probe sequences are listed in Table 1. TaKaRa Premix Ex Taq (Takara Bio, Inc., Otsu, Japan), with a final reaction volume of 20 μL, was used for the TaqMan probe-based quantitative reverse transcription PCR reaction, which was performed on an Applied Biosystems 7500 fast real-time PCR System (Applied Biosystems; Thermo Fisher Scientific, Inc.). The level of mRNA expression in each sample was quantified using the absolute quantification standard curve method. The plasmid cDNA of each gene was used to prepare the absolute standards. The concentration was measured using the A260/A280 ratio, which was converted to the number of copies by using the molecular weight of the DNA. Each mRNA expression level was normalised to that of the housekeeping 18s ribosomal RNA gene.

### 4.6. Mitochondrial Enzyme Activity

To identify abnormal mitochondrial enzyme activity, frozen sections (10 μm thick) of the whole kidney were histochemically stained for COX and SDH enzyme activity. For COX staining, frozen tissue sections were incubated in buffer containing 0.05% tris HCL, 4% sucrose, and 0.06% 3,3′-diaminobenzidine (Sigma-Aldrich) for 2 h at 37 °C. For SDH staining, frozen tissue sections were incubated with a buffer containing 0.5 mg/mL nitro-blue tetrazolium (Sigma-Aldrich) and 50 mmol/L sodium succinate for 1 h at 37 °C. The percentage of positive staining for COX was quantified using a colour image analyser (Keyence).

### 4.7. Senescence-Associated β-Galactosidase Activity

To confirm SA β-gal activity, frozen sections (10 μm thick) of whole kidneys were stained using a Senescence β-Galactosidase Staining Kit (Cell Signaling Technology, Inc.). The percentage of positive staining for SA β-gal was quantified using a colour image analyser (Keyence).

### 4.8. Cell Culture

BMDMs were used for the in vitro assays. Bone marrow was isolated from the femur bone and bone marrow cells. Cells were incubated at 37 °C in an atmosphere containing 5% CO_2_ and differentiated to macrophages using a 20% L929-conditioned medium containing the macrophage colony-stimulating factor for 6 days. To activate the NLRP3 inflammasome, BMDMs were primed for 3 h with ultrapure LPS (InvivoGen, San Diego, CA, USA), followed by stimulation with ATP (5 mM) 30 min before the cell lysates and supernatants were harvested. GSNO (500 μM, Sigma-Aldrich) was treated at the same time as LPS. IL-1β and IL-6 in supernatants were detected using ELISA (Thermo Fisher Scientific, Inc.).

### 4.9. Statistical Analyses

Statistical analyses were performed using GraphPad Prism7 software (GraphPad Software, Inc., San Diego, CA, USA). Comparisons between multiple groups with normal distribution were performed using one-way ANOVA, followed by Tukey’s multiple comparison test, whereas the Kruskal–Wallis test, followed by Dunn’s multiple comparison test, was used for groups with non-normal distribution. All values are expressed as the mean ± SD. Statistical significance was set at *p* < 0.05.

## 5. Conclusions

Overall, our findings revealed the importance of inflammasome activation and disruption of the eNOS-NO pathway in ageing kidneys. Lack of eNOS-NO pathway exacerbated age-related glomerular sclerosis and tubulointerstitial damage. Mitochondrial damage was observed in the proximal tubules of aged mice, which was alleviated by *ASC* gene deletion. NO suppress NLRP3 Inflammasome activation in macrophages directly. Therefore, regulation of inflammasome activation may reduce the progression of age-related renal injury. The importance of endothelial function as a regulatory mechanism suggests that protection of endothelial function may be a potential therapeutic target.

## Figures and Tables

**Figure 1 ijms-22-09269-f001:**
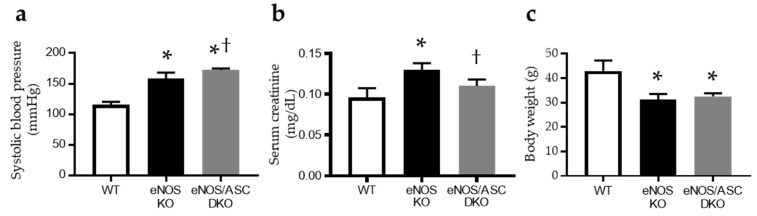
Evaluation of systolic blood pressure and serum creatinine. Evaluation of (**a**) systolic blood pressure, (**b**) serum creatine and (**c**) body weight. Data are expressed as means ± SD. WT, wild type (*n* = 5); eNOS KO, eNOS KO mice (*n* = 4); eNOS/ASC DKO, eNOS/ASC DKO mice (*n* = 4). * *p* < 0.05 vs. WT, † < 0.05 vs. eNOS KO.

**Figure 2 ijms-22-09269-f002:**
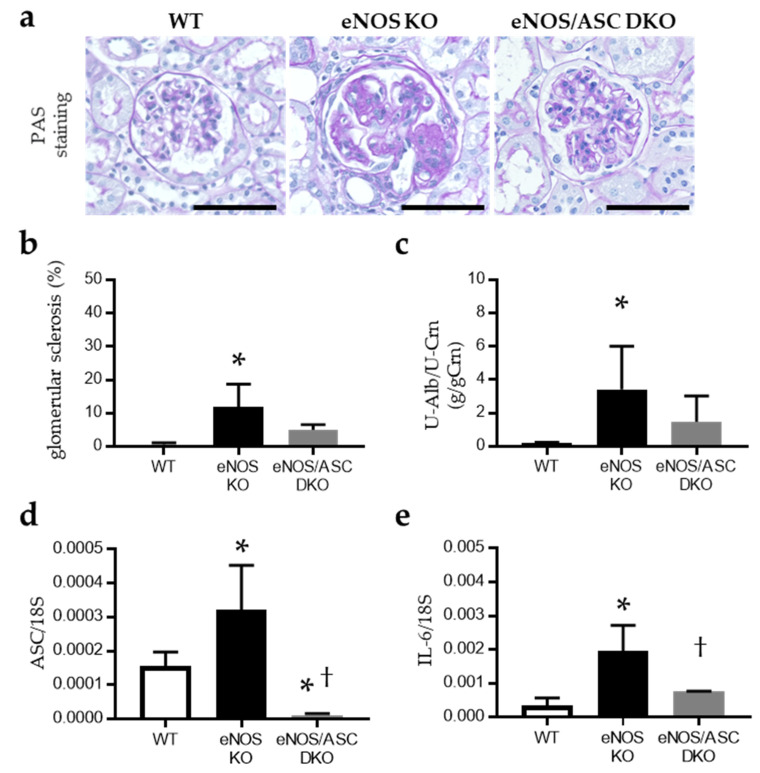
Evaluation of glomerular damage and inflammatory response in aged mice: (**a**) PAS staining of glomeruli. Bar = 50 μm; (**b**) percentage of glomerular sclerosis; (**c**) measurement of U-Alb/U-Crn. mRNA expression of (**d**) ASC, (**e**) IL-6, (**f**) IL-18, (**g**) WISP1 and (**h**) CyclinD1. Data are expressed as means ± SD. WT, wild type (*n* = 3–5); eNOS KO, eNOS KO mice (*n* = 3–5); eNOS/ASC DKO, eNOS/ASC DKO mice (*n* = 3–5); PAS, periodic acid–Schiff; U-Alb/U-Crn, urinary albumin/urinary creatinine. * *p* < 0.05 vs. WT, † *p* < 0.05 vs. eNOS KO.

**Figure 3 ijms-22-09269-f003:**
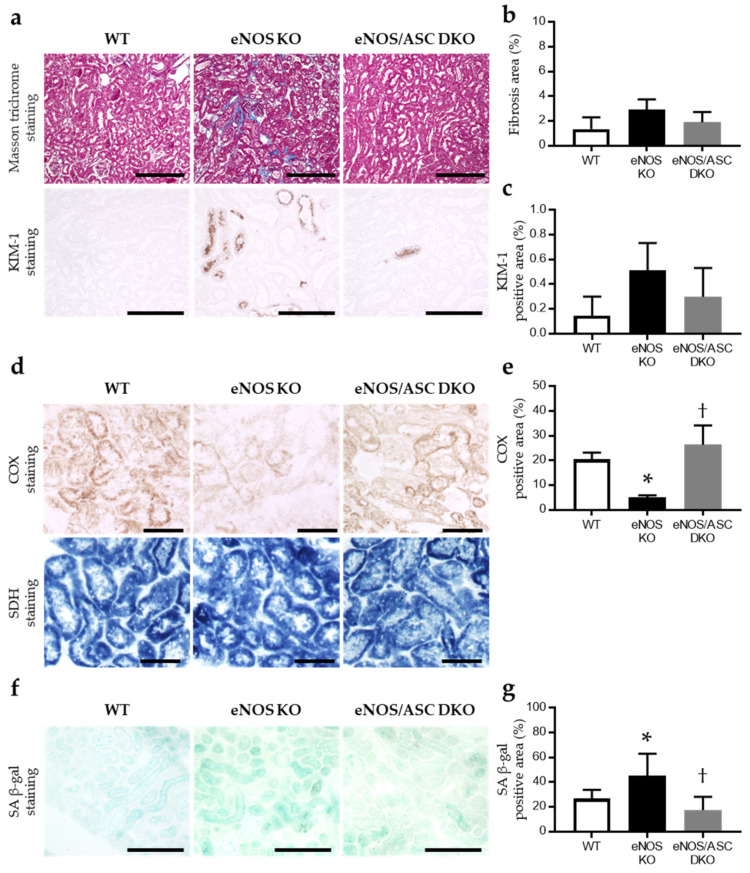
Aged-eNOS KO accelerated mitochondrial damage and cellular senescence but not eNOS/ASC DKO: (**a**) upper panel; Masson trichrome staining (bar = 250 μm), lower panel; immunohistochemical staining for KIM-1 (bar = 100 μm); (**b**) percentage of aniline blue-positive area in Masson trichrome staining; (**c**) percentage of KIM-1 positive area; (**d**) upper panel is COX activity staining (bar = 50 μm); lower panel is SDH activity staining (bar = 50 μm); (**e**) percentage of COX positive area. (**f**) SA β-gal activity staining. Bar = 250 μm; (**g**) percentage of SA β-gal positive area. Data are expressed as means ± SD. WT, wild type (*n* = 3–6); eNOS KO, eNOS KO mice (*n* = 3–6); eNOS/ASC DKO, eNOS/ASC DKO mice (*n* = 3–6); KIM-1, kidney injury molecule 1; COX, cytochrome c oxidase; SDH, succinate dehydrogenase; SA β-gal, senescence-associated beta-galactosidase. * *p* < 0.05 vs. WT, † < 0.05 vs. eNOS KO.

**Figure 4 ijms-22-09269-f004:**
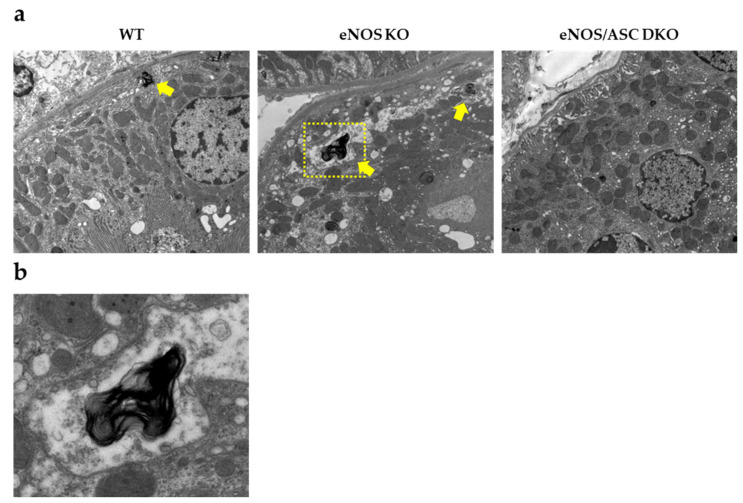
Aged-eNOS KO mice exhibited mitochondrial dysfunction in the proximal tubules: (**a**) transmission electron microscopy image of proximal tubular epithelial cells. Yellow arrows indicate hypertrophied MLBs; (**b**) zoomed-in images of the yellow square, demonstrating the whirl-like structure. Yellow arrows indicate hypertrophied MLBs. MLBs, multilamellar bodies.

**Figure 5 ijms-22-09269-f005:**
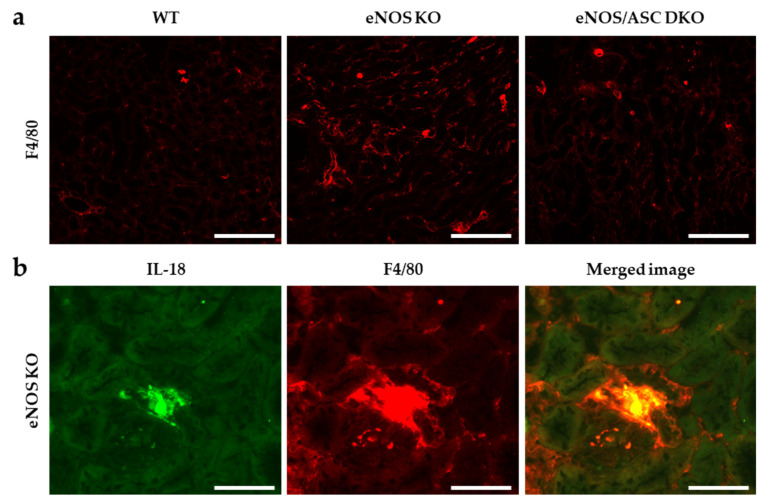
Macrophage infiltration and inflammasome activation in the tissue of aged mice: (**a**) immunofluorescence staining of F4/80 in kidney tissue. Bar = 250 μm; (**b**) double-immunofluorescence analysis of IL-18, F4/80, and the merged image. Bar = 50 μm. Green indicates FITC; red indicates Alexa Fluor 555.

**Figure 6 ijms-22-09269-f006:**
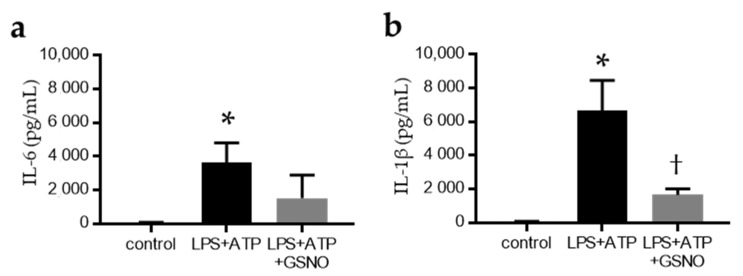
NO directly regulates inflammasome-related cytokines in primary cultured macrophages. Inflammasome activation was evaluated in primary bone marrow-derived macrophages (BMDMs). The levels of (**a**) IL-6 and (**b**) IL-1β were detected with an enzyme-linked immunosorbent assay in the supernatant. Data are expressed as means ± SD. Control, BMDMs without stimulation (*n* = 3); LPS, lipopolysaccharide; LPS + ATP, stimulated with ATP after LPS priming (*n* = 3); LPS + ATP + GSNO, treated with NO donor, GSNO (*n* = 3). * *p* < 0.05 vs. WT, † *p* < 0.05 vs. eNOS KO.

**Figure 7 ijms-22-09269-f007:**
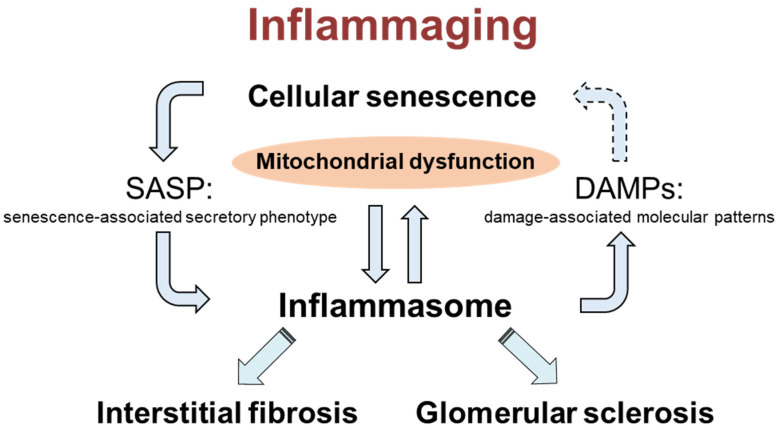
Scheme of the role of inflammasomes in ageing kidneys.

**Table 1 ijms-22-09269-t001:** Primer and probe sequences used for real-time polymerase chain reaction.

Gene	Accession Number	Primer and TaqMan Probe Sequences (5′–3′)
*ASC*	NM_023258	Forward primer: CACCAGCCAAGACAAGATGA
		Reverse primer: CTCCAGGTCCATCACCAAGT
		TaqMan probe: FAM-CCCTCCTCCAGGCCTTGAAGGA-TAMRA
*IL-6*	NM_03168	Forward primer: CTTCACAAGTCCGGAGAGGA
		Reverse primer: TCCACGATTTCCCAGAGAAC
		TaqMan probe: FAM-CAGAGGATACCACTCCCAACAGACCTG-TAMRA
*IL-18*	NM_008360	Forward primer: AGACAGCCTGTGTTCGAGGA
		Reverse primer: AGAGGGTCACAGCCAGTCC
		TaqMan probe: FAM-CAAAGTGCCAGTGAACCCCAGACCA-TAMRA
*WISP1*	NM_018865	Forward primer: ACACATCAAGGCAGGGAAGA
		Reverse primer: CGCAGTACTTGGGTCGGTAG
		TaqMan probe: FAM-CAGCCAGAGGAGGCCACGAACTT-TAMRA
*cyclinD1*	NM_007631	Forward primer: CCTGCTGGAGAAGGTTTAGG
		Reverse primer: ATTGGGTTGGGAAAGTCAAG
		TaqMan probe: FAM-ATTGGTCTTTCATTGGGCAACGG-TAMRA
*18S rRNA*	NR_003278	Forward primer: CCTGCGGCTTAATTTGACTC
		Reverse primer: GACAAATCGCTCCACCAACT
		TaqMan probe: FAM-TCTTTCTCGATTCCGTGGGTGGTG-TAMRA
FAM, 6-carboxyfluorescein; TAMRA, *N*,*N*,*N*′,*N*′-tetramethyl-6-carboxyrhodamine derivative.		
Gene, genes used for qPCR; Accession number, locus of mRNA; Primer and TaqMan probe sequences, sequences of primer and probe.		

## Data Availability

The data presented in this study are available in the article.

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
