# Peer review of "Endothelial Dysfunction Accelerates Impairment of Mitochondrial Function in Ageing Kidneys via Inflammasome Activation"

_ijms, 2021, doi:10.3390/ijms22179269_

Round 1
Reviewer 1 Report
In the paper “Endothelial dysfunction accelerates impairment of mitochondrial function in ageing kidneys via inflammasome activation” the authors aimed to explore the role of ED in aged kidneys. To do this the authors performed experiments in eNOS and ASC null mice. I find the paper very interesting and well written. It is also well designed and with clear methodology. I would just recommend to expand the introduction or discussion part regarding the role of eNOS in endothelial dysfunction in kidney impairment. I would also suggest to take information for this new paragraph and cite a recent review article in this area (doi 10.3390/jcm9082359)
Author Response
Reviewer #1:
In the paper “Endothelial dysfunction accelerates impairment of mitochondrial function in ageing kidneys via inflammasome activation” the authors aimed to explore the role of ED in aged kidneys. To do this the authors performed experiments in eNOS and ASC null mice. I find the paper very interesting and well written. It is also well designed and with clear methodology. I would just recommend to expand the introduction or discussion part regarding the role of eNOS in endothelial dysfunction in kidney impairment. I would also suggest to take information for this new paragraph and cite a recent review article in this area (doi 10.3390/jcm9082359)
→Thank you for you great suggestion and comments. We would like to add discussion about eNOS in endothelial dysfunction in kidney diseases. We added following sentence in line 40.
“Endothelial dysfunction which assessed by Flow-medicated vasodilatation is observed in CKD patients. One of the mechanism of impairment of eNOS-NO pathway is production of reactive oxidative species (ROS). It is well known that ROS are another critical determinant of the bioavailability of NO. Moreover, ROS scavenge NO, resulting in the generation of peroxynitrite, which further decreases NO bioavailability”
Reviewer 2 Report
Three authors of the present work are also authors from papers cited, they are Nagasu H. with 8 auto-citations in the references: 1, 6, 7, 18, 26, 30, 33, 34; the author Kadoya H. with the references 5, 6, 30 and the author Kidokoro K. with the references 1, 6, 7, 30, 33, 34. This results in 26% of auto-citation in a total of 34 references, which is a very high level of auto-citation.
Line 38, line 201, etc. "We have reported..."; "our results..." - Science is written in the third person, it should not be personalised. Please, fix all the text regarding this issue.
The authors works cited by the name in the discussion text are only from the authors of the present work, which could reveal that author's knowledge, about the state of the art regarding to the issues discussed is sparse.
Statistical analyses - Given the number of data, this work should use mean ± SD and not mean ± SEM.
The number of events for data acquisition and the number of replication must be included in each and every one method described.
Line 179 - "However, since this study used general ASC KO mice, we could not examine which cells are important for inflammasome activation in the ageing kidney." - A better discussion about the choice of the mice model used is needed.
A better explanation about the mice models used are important for the readers understanding.
Author Response
Reviewer #2:
Three authors of the present work are also authors from papers cited, they are Nagasu H. with 8 auto-citations in the references: 1, 6, 7, 18, 26, 30, 33, 34; the author Kadoya H. with the references 5, 6, 30 and the author Kidokoro K. with the references 1, 6, 7, 30, 33, 34. This results in 26% of auto-citation in a total of 34 references, which is a very high level of auto-citation.
→We apologized for this issue. We decided to delete ref6, 17 and 33. Then we added new ref33.
Line 38, line 201, etc. "We have reported..."; "our results..." - Science is written in the third person, it should not be personalised. Please, fix all the text regarding this issue.
→ Thank you for your comments. I agree with you totally. We corrected these sentences. Please see line 37,173.
The authors works cited by the name in the discussion text are only from the authors of the present work, which could reveal that author's knowledge, about the state of the art regarding to the issues discussed is sparse.
→ Thank for your great suggestion. We changed the sentence line 173,188.
Statistical analyses - Given the number of data, this work should use mean ± SD and not mean ± SEM.
→ We are sorry to lack of number in method section. We added the number of mice in method section and change from SEM to SD.
The number of events for data acquisition and the number of replication must be included in each and every one method described.
→ We apologize for lacking information about number of experiments. We added these information in figure legends of figure6.
Line 179 - "However, since this study used general ASC KO mice, we could not examine which cells are important for inflammasome activation in the ageing kidney." - A better discussion about the choice of the mice model used is needed. A better explanation about the mice models used are important for the readers understanding.
→ It is very important point for understand the role of inflammasome in aging kidney. _we added following sentence in line 195.
“In addition, there are some reports of inflammasome activation in tubular cells. In the future, it will be necessary to use cell-specific genetically modified animals to clarify this issue. Or bone marrow transplantation experiments could help to dissolve this issue.”
Thank you again for your kind comments and suggestion. I believe that this revised manuscript could be suitable for publication on Internal Journal of Molecular Sciences.
Round 2
Reviewer 2 Report
"Line 38, line 201, etc. "We have reported..."; "our results..." - Science is written in the third person, it should not be personalised. Please, fix all the text regarding this issue.
→ Thank you for your comments. I agree with you totally. We corrected these sentences. Please see line 37,173":
Please, correct all the text regarding this issue. As line 14, 16, 140, 169, 176, 185, 207, 219, 227, ETC.
Nagasu H. : 6 auto-citations - References: 1, 8, 18, 26, 32
Kadoya H.: 2 auto-citations - References: 7, 34
Kidokoro K.: 3 auto-citations - References:1, 8, 32
The auto-citaton level is still high for a paper with just 34 references.
"The number of events for data acquisition and the number of replication must be included in each and every one method described.
→ We apologize for lacking information about number of experiments. We added these information in figure legends of figure6.":
This is the kind of information that must be present in the methodology description. Plus "Are the methods adequately described?
( ) ( ) (x) ( ).
The future perspectives and the potential application of these findings are still needed in the end of the abstract and in the conclusion.
Figure 3g) - Please, include the 100% mark in the YY axis. The 1 in the c); 50% in the e) - Regarding the previous "Are the results clearly presented?
( ) ( ) (x) ( )". This issue should be considered in all the results presented.
See all topics carefully, please.
"The authors works cited by the name in the discussion text are only from the authors of the present work, which could reveal that author's knowledge, about the state of the art regarding to the issues discussed is sparse.
→ Thank for your great suggestion. We changed the sentence line 173,188." :
The idea is not to only delete the name of the authors from the text, but also to add more discussion and more information regarding other authors work and increase the backing for this topic. If the literature has no data about this, then this should also be discussed.
The weight of the animals is still lacking, as well as the gender, and a table or scheme about in vivo protocol. - Regarding: "Are the methods adequately described? ( ) ( ) (x) ( )."
What about "Are the conclusions supported by the results?
( ) (x) ( ) ( )". It is still the exact same conclusion.
More rigor is needed in the writing and presentation of these scientific results. In addition, the first round of peer-review has to be seen with more rigor and certainty. When peer-review writes "etc", it means that more than the lines identified by the reviewer have to be seen in that context by the authors.
The statistical treatment of results with SEM in version V1 of the document and with SD in version V2 does not change the results of Figure 1 a) and b) at all, only the legend was changed. In figure 2, the scale of the graph b), f), g) and h) was changed. Mathematics is not only reflected in the figure's caption. The same happens in the graphs and legend of figure 3, only the graph had a change in the scale of the Y axis, from 60 to 80 of maximum value. In Figure 6, there is a change in the legend and scale of a) on the Y axis from 5000 to 6000. Mean +/- SEM and mean +/- SD are numerically different which is objectively and measurably reflected graphically. Where is the real statistic treatment suggested by the reviewer?
Round 3
Reviewer 2 Report
Please, provide the weight of used animals in the materials and methods section (4. Materials and Methods, Line 276: 4.1. Animal).
The conclusion needs more details about the achievements of the present work.
Aside from these, thank you to the authors for taking into consideration this reviewer's comments.
